# Contribution of Adventitia-Derived Stem and Progenitor Cells to New Vessel Formation in Tumors

**DOI:** 10.3390/cells10071719

**Published:** 2021-07-07

**Authors:** Berin Upcin, Erik Henke, Florian Kleefeldt, Helene Hoffmann, Andreas Rosenwald, Ster Irmak-Sav, Huseyin Bertal Aktas, Uwe Rückschloß, Süleyman Ergün

**Affiliations:** 1Institute of Anatomy and Cell Biology, Julius-Maximilians-University, 97070 Würzburg, Germany; berin.upcin@uni-wuerzburg.de (B.U.); erik.henke@uni-wuerzburg.de (E.H.); florian.kleefeldt@uni-wuerzburg.de (F.K.); helene_hoffmann@gmx.net (H.H.); uwe.rueckschloss@uni-wuerzburg.de (U.R.); 2Institute of Pathology, Julius-Maximilians-University, 97070 Würzburg, Germany; rosenwald@uni-wuerzburg.de; 3Faculty of Health Sciences, İstanbul Bilgi University, 34060 Istanbul, Turkey; ster.irmak@bilgi.edu.tr; 4Department of Medicine, Hematology, Brigham and Women’s Hospital, Boston, MA 02115, USA; huseyin_aktas@hms.harvard.edu

**Keywords:** vascularization model, tumor spheroids, vascular wall stem and progenitor cells, aortic adventitia, vasculogenesis, tumor-vessel wall-interface model

## Abstract

Blocking tumor vascularization has not yet come to fruition to the extent it was hoped for, as angiogenesis inhibitors have shown only partial success in the clinic. We hypothesized that under-appreciated vascular wall-resident stem and progenitor cells (VW-SPCs) might be involved in tumor vascularization and influence effectiveness of anti-angiogenic therapy. Indeed, in patient samples, we observed that vascular adventitia-resident CD34^+^ VW-SPCs are recruited to tumors in situ from co-opted vessels. To elucidate this in detail, we established an ex vivo model using concomitant embedding of multi-cellular tumor spheroids (MCTS) and mouse aortic rings (ARs) into collagen gels, similar to the so-called aortic ring assay (ARA). Moreover, ARA was modified by removing the ARs’ adventitia that harbors VW-SPCs. Thus, this model enabled distinguishing the contribution of VW-SPCs from that of mature endothelial cells (ECs) to new vessel formation. Our results show that the formation of capillary-like sprouts is considerably delayed, and their number and network formation were significantly reduced by removing the adventitia. Substituting iPSC-derived neural spheroids for MCTS resulted in distinct sprouting patterns that were also strongly influenced by the presence or absence of VW-SPCs, also underlying the involvement of these cells in non-pathological vascularization. Our data suggest that more comprehensive approaches are needed in order to block all of the mechanisms contributing to tumor vascularization.

## 1. Introduction

In order to sustain their continuous growth, tumors have to secure an adequate supply of nutrients and oxygen by inducing vascularization processes. Tumors that fail to acquire the ability to initiate the formation of a vascular supply remain in the status of dormant microlesions [1,2]. In general, tumors can secure vascular supply by the following three means: angiogenesis, that is the formation of new blood vessels from existing ones; vasculogenesis, i.e., the construction of neovessels from progenitor cells [3], which can be activated from reservoirs that reside i.a. in the bone marrow [4,5]; or the co-option of nearby vessels that get incorporated into the tumor tissue by its expansive growth. Under most conditions, and especially under those that tend to get evaluated by researchers, the main process by which tumor blood vessels are formed appears to be sprouting angiogenesis. However, under certain conditions, other forms of vascularization can significantly contribute to maintaining tumor supply; Paez-Ribes et al. have shown that when angiogenesis is blocked, glioma cells invade into the surrounding parenchyma along existing blood vessels under anti-angiogenics, thereby co-opting these vessels for their metabolic needs [6]. Similarly, under therapy-induced acute hypoxic stress, tumors secrete high levels of cytokines like VEGF-A and SDF-1 that mobilize endothelial progenitor cells (EPCs) from the bone marrow [7]. This vasculogenic rescue restores, in parts, tumor perfusion faster than possible with angiogenic sprouting. The question of how tumors get vascularized is not just an academic one. Distinctive molecular mechanisms are involved in the regulation of the different vascularization processes. Thus, they are also differently susceptible to anti-angiogenic therapies. The contribution of alternative cellular sources to vascularization processes can therefore render anti-angiogenic approaches ineffective, and switching to alternate vascularization pathways can provide an escape mechanism.

Here, we examined the contribution of an often-overlooked cell population to tumor vascularization, namely vascular wall-resident stem cells (VW-SPCs) that reside mainly within the adventitial layer of large and mid-sized blood vessels [8]. To identify the role of VW-SPCs in vascular sprouting and the local generation of macrophages, we established a modified protocol for the aortic ring assay (ARA) that allowed for distinguishing the role of adventitia-derived cells from those of other layers of the vascular wall, particularly from the intimal mature endothelial cells and subendothelial progenitors.

Although neovessels are also generated from the intima layer, progenitor cells from the adventitia of mature adult vessels have been shown to strongly contribute to the formation and morphogenesis of new vascular structures. Importantly, using the modified ARA protocol, we were able to demonstrate that the cells derived from the adventitia play a crucial role in the response of neovessels to therapeutics. We then established an ex vivo 3D tumor−vessel wall interface model, where the aortic rings of C57BL/6-Tg (UBC-GFP) mice and MCTS were confronted in ex vivo culturing. With this model, it was possible to observe the mobilization of VW-SPCs by and their recruitment to the tumor, and to thus follow their contribution to tumor vascularization in real time. Moreover, it enabled the differential evaluation of the different tumor vascularization mechanisms and their response to varied culture conditions and treatment.

## 2. Materials and Methods

### 2.1. Reagents

Standard chemicals and reagents, if not stated otherwise, were purchased from SigmaAldrich (Munich, Germany). VEGF-A (PN# 450-32) was purchased from Peprotech (Hamburg, Germany). ZM323881 (PN# S2896) was purchased from Selleckchem (Munich, Germany). Axitinib (PZ0193) was purchased from SigmaAldrich (Munich, Germany). Rat collagen type I (PN# 08-115) was from Millipore (Darmstadt, Germany) and Collagenase II (PN# LS004176) was purchased from Worthington (Lakewood, CA, USA). Opti-MEM I (1x)-GlutaMAX™-I (51985-026) was purchased from Thermo Fisher Scientific.

### 2.2. Antibodies

The following primary antibodies were used: F4/80 (Abcam, ab16911, Berlin, Germany), CD34 (Abcam, ab8158, Berlin, Germany), NG2 (Merck Millipore, AB5320, Darmstadt), and CAIX (Santa Cruz, sc-365900, Heidelberg, Germany).

For DAB staining, biotinylated secondary antibodies from Vector Laboratories (Eching, Germany) were used (anti-rat IgG BA-9400). For fluorescence staining, secondary antibodies from Jackson ImmunoResearch (Cambridge, UK) were used (Cy3-conjugated AffiniPure goat anti-rat (112-165-003) and Cy5-conjugated AffiniPure goat anti-rabbit IgG 1(11-175-144) from Thermo Fisher Scientific (alexa fluor^®^ plus 555 anti mouse (A32727)).

### 2.3. Aortic Ring Assay (ARA)

Based on the method published by Baker et al. [9], we developed a modified ARA (mdARA) protocol for high inter-assay reproducibility. Like Baker et al., we found that the reproducibility of the ARA is highly dependent on the type of collagen used for embedding ARs, and only type I rat tail collagen resulted in consistent and reproducible sprouting. Likewise, the use of the 96-well format was proven to be critical in our hands. Particular care has to be taken to not exert increased pressure on the intima of the aorta. Therefore, our protocol refrains from any perfusion steps. After dissection of the aorta, the remaining blood cells were washed out by careful agitation of the separated rings in the medium after cutting the aorta. Under a dissection microscope, the aorta was cut into rings of ~0.7–1.0 mm in thickness with a scalpel. The rings were transferred to a 10 cm petri dish with 3 mL Opti-MEM without serum and growth factors, but supplemented with 1% (*v*/*v*) penicillin. ARs were starved overnight at 37 °C in a standard incubator. After overnight starvation, the ARs with (Adv(+)-ARs) or without adventitia (Adv(−)-ARs)) were embedded with or without MCTS in collagen. To improve the success rate of the assay, we increased the volume to 80 µL collagen I per well. The rat collagen solution was prepared as described by Baker et al. [9].

First, 150 µL Opti-MEM with 2.5% FCS (heat-inactivated) and VEGF-A 30 ng/mL supplemented with 1% (*v/v*) penicillin-streptomycin was added to the collagen embedded ARs. Plates with the embedded ARs were placed in a standard incubator at 37 °C supplemented with 5% CO_2_. Every three days the medium was changed, and images were captured with phase-contrast microscopy for the quantitative analyses. To evaluate the effect of hypoxia on ARs, the embedded ARs were first cultivated for 72 h under normoxic conditions (20% O_2_) in a standard incubator before switching to hypoxic conditions (2% O_2_) for an additional 8–11 days (depending on the experiment) by transferring the plates to an incubator with adjustable O_2_ (HeraCell 240i, Thermo Fisher). At termination of the experiment, the medium was removed and the samples were washed once with PBS and fixed with 4% (*w/v*) paraformaldehyde (PFA)/PBS (pH = 7.4) for ON at room temperature (RT). After fixation, the samples were washed with PBS and embedded in the paraffin for immunohistochemistry of the paraffin sections (IHC-P) or were stored in PBS at 4 °C in the dark for whole-mount clearing and staining.

### 2.4. Removal of the Adventitia Layer

For each thoracic aorta, 781.25 U (2.5 mg) collagenase II was (Millipore, Darmstadt, Germany) prepared in 1.5 mL PBS. The aorta was incubated in the collagenase solution for 10 min at 37 °C and then washed gently in PBS. The adventitia was removed by holding the aorta at one end with fine-forceps and with a second pair of fine-forceps the adventitial layer was grasped and pulled off along the aorta. Under a dissection microscope, the stripped aorta was cut into rings of ~0.7–1.0 mm in thickness with a scalpel.

### 2.5. AR-MCTS Confrontation

MCTS were generated from MDA-MB 435s (ATCC, HTB-129) tumor cells under non-adherent conditions using a modified liquid overlay technique [10]. In brief, 96-well plates were coated with 50 µL 1% agarose. The tumor cells were seeded at 1000 cells/well in 200 μL in DMEM supplemented with 10% FBS and penicillin/streptomycin. The plates ware incubated at 37 °C and 5% CO_2_ for 9 days with 50% of media exchanged every 3 days.

The ARs were prepared from the freshly dissected aortas of GFP mice (C57BL/6-Tg (UBC-GFP), as described above. First, a segment of a 20µL pipette tip was cut with a sterile scalpel. MCTS were absorbed using a carefully cut tip and as little liquid as possible was placed in the middle of a new 96 well plate, 80 µL collagen mixture added to the spheroids, and ARs (Adv(+)-ARs or Adv(−)-ARs) were immediately placed close to the spheroid. To let the collagen solidify, plates were placed at 37 °C in a standard incubator before the ARA medium was added. To evaluate the effect of hypoxia, the plates were cultivated for 72 h under normoxic conditions (20% O_2_) in a standard incubator before switching to hypoxic conditions (2% O_2_) for an additional 8–11 days (depending on the experiment) by transferring the plates to an incubator with adjustable O_2_ (Marke, Model). The samples were fixed with 200 μL/well of 4% (*w/v*) PFAin PBS ON at RT.

### 2.6. Immunohistochemistry and Immunofluorescence

Paraffin embedded tissue sections were deparaffinized in Xylol and rehydrated using a descending alcohol series. To block the endogenous peroxidase activity, sections were incubated in 3% (*v/v*) H_2_O_2_ in dH_2_O for 10 min. Demasking was performed either under acidic (10 mM sodium-citrate, pH = 6.0) or basic conditions (10 mM Tris plus 1 mM EDTA, pH = 9.0) for 30 min at 95 °C. Sections were blocked in 5% NGS/PBS for 1 h and were incubated with primary antibodies over night at 4 °C. Antigen was detected with a peroxidase-conjugated secondary antibody (1/250) and DAB staining (Dako). The samples were gradually dehydrated via consecutive incubations in ethanol/xylene and mounted with DePex from Serva and a coverslip. For the immunofluorescence analysis, the antigen was detected with an anti-rat Alexa Cy3 and anti-rabbit Alexa Cy5- conjugated secondary antibody (1/400). DAPI was used for nuclei staining.

### 2.7. Whole Mount Staining (Clearing)

The fixative was removed from the fixed and collagen-embedded samples in the 96-well plates. The samples were washed with PBS before placing them in 2 mL microcentrifuge tubes and were incubated in 500 μL of penetration buffer (0.25% Triton X-100 and 25% glycin in PBS) for 1 h at RT. The samples were then incubated in a blocking buffer (0.25% Triton X-100, 6% BSA, 10% DMSO in PBS) for 1 h at RT and washed twice for 1 h in a washing buffer (0.2% Tween-20 in PBS) at RT. The primary antibodies were applied in 100 µL of antibody buffer (0.2% Tween-20, 3% BSA, 5% DMSO in PBS) ON at 4 °C. The samples were then washed 6× for 30 min with a washing buffer at RT. The samples were incubated ON at 4 °C in the dark with secondary antibodies in antibody buffer. After the incubations time, the samples were washed with washing buffers for 6× 30 min at RT in the dark. Staining was concluded by incubation in a DAPI solution for 30 min at RT and washes for 8× 20 min with washing buffer at RT. The samples were dehydrated in an ascending EtOH series of 1× 50%, 1× 70%, 1× 90%, and 3× 100% EtOH in 10 mM TrisHCl pH 9.0 for 1 h at RT for each step. The collagen plugs were finally transferred to a glass vial and cleared by incubation in two changes of ethyl cinnamate for 12 h at RT. Images were acquired on a Nikon A1 confocal microscope using a Nikon Plan Apo 2/0.75, 60×/1.40 objective.

### 2.8. Quantification

The sprouting length, sprouting area, number of sprouts, and number of branches were quantified during the experiment (nonfixed, live rings) by live phase-contrast microscopy. The sprouting length was quantified as the mean maximal sprout from the perimeter of the ARs to the most distal tip of the angiogenic sprout in four quadrants of each AR, and the sprouting area of each AR was measured by marking the total area of all of the cells, and the number of sprouts was quantified as the mean counted sprout in four quadrants of each AR. The number of branches was calculated as the mean total length of the capillary divided by the mean of length of the distance from branching (branch length). The average branch length was quantified as the mean total length of the capillary divided by the mean number of branches. Per treatment condition, six ARs were analyzed. For each AR, eight branch lengths were measured for four quadrants. Experiments were repeated at least three times. The area of the migrated GFP^+^ aortic cells into spheroids was quantified by measuring the total cell area using ImageJ program. All of the measurements were performed using ImageJ software (https://imagej.nih.gov/ij/). Error bars indicate SDs. *p* values were calculated by unpaired Student’s *t*-test.

IHC staining was also quantified using ImageJ. The same threshold parameters were used for the quantification of all of the images in a series.

### 2.9. Statistical Analysis

All of the statistical analyses were done using Prism5 Software (GraphPad, LaJolla, CA, USA). The differences between the two groups were analyzed using an unpaired, two-tailed Student’s *t*-test. In parallel, the samples were tested for a significant variation of variance, and if necessary, a Welch correction was included in the statistical analysis.

## 3. Results

### 3.1. Mobilization of Adventitia-Resident CD34^+^ Cells in Human Tumors In-Situ

As a first step to investigate the role of adventitial cells on tumor vascularization, we examined histological sections of human tumors by immunohistological staining for CD34, a marker expressed in various hematological stem cell populations, but also in endothelial progenitors and mature endothelial cells [7,11]. Moreover, CD34 is a reliable marker for monitoring adventitial VW-SPCs, as the majority of these cells express CD34 but down-regulate it to some extent when they detach from the adventitial niche [8,12]. We hypothesized that with these properties, CD34 might also be a suitable marker to evaluate the contribution of VW-SPCs to the vascularization of human tumor tissues in situ. Accordingly, we first stained human bladder cancer sections for CD34, and examined various areas in the surrounding normal part, the periphery of the tumor, and within the tumor mass of the sections. These analyses revealed that CD34^+^ cells are gradually mobilized from the adventitia of pre-existing peritumoral blood vessels (BVs), an effect that gradually gets stronger with the closer proximity of BVs: in remote tumor-free zones, blood vessels displayed a strong CD34 staining in the entire circumference of the vascular adventitia (Figure 1A). In close proximity to the tumor area, CD34^+^ cells are mobilized from the adventitia and are recruited to the tumor tissue in the form of cell cords. These cell cords either surround the tumor cell islands (Figure 1B) or penetrate into the tumor tissue to form new vessels (Figure 1B,C). In areas surrounding the tumor without visible invasion of tumor cell clusters, the adventitia of blood vessels still displays CD34 staining in almost the entire adventitia circumference, as in normal blood vessels (Figure 1D). In pre-existing BVs completely co-opted by the tumor, the adventitia often appeared to be entirely free of CD34^+^ cells, while the media layer of the vessel wall composed mainly of smooth muscle cells remained largely intact (Figure 1E). The intimal endothelial cells lining of the lumen of these BVs changed from the original flat structure into a more rounded shape (Figure 1F), probably because of the altered endothelial layer organization caused by tumor-secreted factors. By staining human breast cancer sections for CD34, we corroborated the results from the bladder cancer samples (Appendix A). These results pose the question of the role and the fate of the mobilized CD34^+^ adventitial cells in tumor vascularization. As this is difficult to study in vivo, we decided to utilize a vascular explant model based on the aortic ring assay (ARA).

### 3.2. Differential Testing of Capillary Sprouting Capacity of VW-SPCs in a Modified Aortic Ring Assay (mdARA)

To examine the contribution of VW-SPCs to new vessel formation, e.g., in the context of tumor vascularization, we modified the conventional ARA procedure by establishing a protocol that enabled the secure removal of the entire adventitial layer prior to embedding, without compromising the aortic intima. Examination of the H&E-stained sections confirmed both the complete removal of the adventitia and the preserved integrity of the aortic media and intima (Figure 2A,B). Furthermore, even after the removal of the adventitia, we could observe the activation of intimal endothelial cells (ECs). Intimal ECs formed capillary-like sprouts into the lumen of the aortic rings, independent of prior removal of the adventitia (Figure 2C,D). Closer examination revealed that explants showed massive mobilization of individual cells into the collagen matrix. The majority of these cells did to not directly contribute to the formation of capillary-like sprouts. This effect was much stronger when the adventitia was left in place. A comparison of the sprouting behavior and pattern of ARs embedded into collagen with and without the adventitia allowed for studying the contribution of the adventitia-derived VW-SPCs under various treatment modalities. Basic cultivation of the ARs with adventitia resulted in an extensive capillary-like sprouting and a dense network formation within the collagen gel (Figure 2E). In contrast, after the removal of the adventitia, cultivation of the embedded ARs still resulted in solid capillary-like sprouting (Figure 2F), but the sprouting pattern was significantly rarefied, as the number of sprouts and the complex network formation were considerably reduced (Figure 2F). Remarkably, the radial length of the capillary-like sprouts was considerably longer in the ARs without adventitia in place (Figure 2E,F).

### 3.3. Influence of Adventitia-Derived VW-SPCs on Capillary-Like Sprouting

We next examined the differences in the sprouting behavior and patterns in more detail—while the ARs with adventitia in place displayed sprouting cells within the collagen gel already after 24 h, this was observed for ARs without adventitia starting after only the third day of culture. Consistently, ARs with an adventitial layer exhibited capillary-like sprouting within the first 2 days of culture (Figure 2G,H), whereas ARs without adventitia displayed such morphogenetic events later—on culture days 3–4 (Figure 2G’–I’). During further cultivation, the single individual capillary sprouts from the ARs without adventitia grew distinctively radially from the AR (Figure 2J’). Quantitative evaluation of the sprouting area confirmed the results and demonstrated a significantly earlier sprouting and significantly larger area occupied by the sprouting cells and capillaries when the adventitia was left in place (Figure 2K). Accordingly, the sprouting density of the capillary-like structures was significantly higher in the ARs with adventitia than in those without adventitia (Figure 2L). In contrast, the maximum sprouting length achieved by individual capillary-like sprouts was considerably longer when they were generated without the assistance of adventitia-derived VW-SPCs (Figure 2M). Next, we evaluated the sprouting network formation by counting the branching points of the individual sprouts, and these analyses showed a significantly higher branching index in ARAs with adventitia compared with those without (Figure 2N), while the fewer individual branches were significantly longer in the ARAs without adventitia (Figure 2O).

Together, these observations supported the hypothesis that besides providing new vessel-forming cells, the adventitia contribute to the sprouting process by supplying both the pro-angiogenic factors, which result in a more complex and enhanced network formation, and cells, which supported the stabilization of these structures.

### 3.4. The ROLE of Vascular Adventitia on Sprouting

To further explore the role of the adventitia on vascular sprouting, we studied the immunophenotype of the cells involved in the sprout morphogenesis in ARs with and without adventitia. These analyses revealed that sprouts from ARs with adventitia left in place display a high number of NG2^+^ cells that are associated with capillary-like sprouts (Figure 3A), while sprouts that were formed in the absence of the adventitia showed significantly reduced coverage by NG2^+^ cells (Figure 3B). Moreover, in the presence of the adventitia, NG2^+^ cells enwrapped the endothelial sprouts from the outside in a pericyte-like manner (Figure 3C). In contrast, the fewer NG2^+^ cells that were detected in the absence of the adventitia showed a more irregular coverage of sprouts, e.g., not sufficiently flattened and not appropriately attached to the endothelial cells (Figure 3D). These data suggest that vascular adventitia-derived VW-SPCs contribute to the generation of NG2^+^ cells, which in turn contribute to the maturation, stabilization, and network formation of capillary-like sprouts (Figure 3E). Besides the NG2^+^ cells in this study, adventitia has been shown to also be a source for CD34^+^ cells (Figure 3F).

As macrophages contribute significantly to vascularization processes under pathological conditions, we next focused on F4/80^+^ cells. In contrast with sections of freshly isolated aorta (Figure 3G), a considerable number of F4/80^+^ cells were found within the entire adventitial layer of ARs after embedding in collagen for seven days (Figure 3H). When the adventitia was removed prior to embedding, F4/80^+^ were not generated in the remaining aortic layers, indicating that vascular adventitia-resident VW-SPCs serve as a crucial source for the generation of local F4/80^+^ macrophage-like cells (Figure 3I). Adventitia-derived F4/80^+^ macrophage-like cells have been shown to express high levels of VEGF-A, the most potent angiogenic and vasculogenic factor [13]. Therefore, we performed double immunofluorescence staining for VEGF and F4/80 on ARA sections with the adventitia left in place. These studies revealed the VEGF-A expression by adventitia-derived F4/80^+^ cells (Figure 3J,K). To better follow the distribution pattern of the macrophages, we used ARs of Cx3cr1 reporter mice in ARA. This allowed for the observation of fluorescent Cx3cr1^+^ cells alongside the capillary-like sprouts (Figure 3L). These findings were corroborated by immunostaining for F4/80 on ARA sections (Figure 3M–O). Furthermore, macrophages were found to be associated to the stem of branching sprouts, and displayed numerous filopodia, suggesting their activation in ARA (Figure 3M,O).

To study the contribution of adventitial cells to capillary morphogenesis, we embedded the isolated adventitial layers from the murine aortas in collagen. Immediately after isolation the adventitial cells resided, inactivated in their niche (Figure 3P). After cultivation for 7 days, capillary-like sprouts were clearly visible within the adventitial tissue (Figure 3R). Immunostaining for NG2 and CD34 demonstrated the complex structure of those sprouts, even without any contribution or support from the cells of the media or intima (Figure 3T). Staining for F4/80 indicated again the vascular adventitia as the sole source for vessel wall-derived macrophages (Figure 3U,V). In conclusion, these experiments showed that VW-SPCs have the potential to generate fully functional vascular sprouts containing both endothelial and peri-vascular cells. Moreover, they also deliver cells, e.g., macrophage-like cells, that contribute to new vessel formation by secreting pro-angiogenic factors. In contrast, the other layers of the vascular wall lack the ability to give rise to supporting cells. Consequently, while the intimal endothelial layer by itself can form capillary-like structures by angiogenic expansion, the formed sprouting networks lacked complexity without the contribution of adventitial-derived cells and factors.

### 3.5. Ex Vivo 3D Tumor–Vessel Wall Interface Model

After demonstrating the contribution of adventitia-derived VW-SPCs to new vascular formation, we established a confrontation model that allowed for studying the contribution of adventitial VW-SPCs to tumor vascularization directly. By co-embedding a multi-cellular tumor spheroid (MCTS) in close proximity to the AR within the collagen gel, we were able to study the interaction of the vascular wall and tumor. ARs were prepared from C57BL/6-Tg (UBC-GFP) mice to distinguish the cells of the AR from those originating from the MCTS. We focused on the following three different regions in the AR-MCTS setup: the interface area directly between the MCTS and the AR, the side of the AR averted from the MCTS (non-tumor side), and finally the side of the MCTS averted from the AR (non-aortic side; Figure 4A,B). When ARs with intact adventitia were confronted with MCTSs, we observed the migration of single GFP^+^ aortic cells into the interface region that also infiltrated the MCTS. In addition to these individual cells, capillary sprouting started in the interface area between AR and MCTS. Interestingly, these analyses demonstrated morphogenesis of strongly truncated capillaries within the tumor−aortic wall interface (Figure 4A,B), while the opposite non-tumor side was characterized by the formation of strong capillary-like sprouts comparable to the sprouts observed in ARA without MCTS (Figure 4C,D). Quantification of the capillary sprouts’ length in both areas confirmed the significantly inhibited capillary-formation in the tumor−vessel wall interface area (Figure 4E and Appendix A). Further analyses with particular focus on the MCTS revealed a higher density of GFP^+^ aortic wall-derived cells within the MCTS under hypoxia in comparison with the normoxic condition (Figure 4F,G). This was also corroborated by the quantification of GFP^+^ cells (Figure 4H). Immunostaining studies on the tissue sections of ARA-MCTS-confrontation showed much less CD34^+^, as well as NG2^+^ cells under normoxia in comparison with hypoxia (Figure 4I,J). Of note, central necrosis in the MCTS was significantly enhanced under normoxia compared with hypoxia. The response to hypoxia was confirmed by IHC-staining of sections for carbonic anhydrase IX (CAIX) (Appendix A).

In the interface area between AR and MCTS, single cell sprouts formed capillary like structures that were maintained until day 5 (Figure 5A), but after day 5 (Figure 5B) they started to disappear, apparently due to their resolution into single cells again. To investigate the differences between the vascularization in the tumor microenvironment and vessel formation in normal tissue, we performed the confrontation culture of AR-MCTS co-culture (Figure 5C,D) parallel to AR with iPS-derived neural spheroids (iNS; Figure 5C). In AR-iNS, the capillary sprouts penetrated directly into the iNS (Figure 5C), while in the AR-MCTS-culture the sprouts remained at the MCTS border (Figure 5D). Next, we wanted to better characterize the relation of adventitia-derived VW-SPCs to MCTS, and thus performed an AR-MCTS-confrontation culture using ARs with and without aortic adventitia in place. In comparison with the co-culturing of MCTS with ARs containing an adventitial layer, where a high number of GFP^+^ cells that originate from the ARs of GFP mice penetrated into the MCTS (Figure 5E) in the co-culturing of MCTS, with ARs without an adventitial layer, only a few GFP^+^ fluorescently labeled cells were observed within the MCTS. The majority of mobilized GFP^+^ cells encircled the MCTS spheroids, covering them from the outside (Figure 5F).

Finally, we examined the effect of pharmacological angiogenesis inhibitors in our tumor−vessel wall interface model. We first used ZM323881, a potent VEGF-R2 inhibitor, which is in contrast to the approved anti-angiogenic TKIs highly selective for VEGF-R2, with a low to negligible affinity for VEGF-R1, -R3, or PDGFRs. Based on the data from the literature, ZM323881 was applied at 4 µM and 10 µM to ARs with the adventitia left in place versus ARs without the adventitial layer [14,15,16]. Treatment with ZM323881 resulted in a clear reduction in the number and length of capillary sprouts under both conditions (Figure 6A–D). However, the effect was considerably stronger when the aortic adventitia was removed (Figure 6E). Remarkably, the sprouting of single cells from the ARs with adventitia was not significantly inhibited under treatment with ZM323881. We then applied ZM323881 in the confrontation culture with MCTS (Figure 6F–I’). In comparison with the corresponding controls (Figure 6F–G´), ZM323881 treatment suppressed the capillary-like sprouting (Figure 6H–I´). Again, we observed a strong sprouting of individual cells into the tumor−aortic wall interface, as well as into the non-tumor side, indicating that a considerable part of the mobilization of cells from the aortic rings is not related to VEGF-R2 signaling. Immunofluorescence staining showed that ZM323881 specifically inhibits the mobilization of CD34^+^ cells, while NG2^+^ cells continued to migrate and also to penetrate into the spheroids (Figure 6J,K). To conclude the evaluation of inhibiting the VEGf pathway, we tested the effect of axitinib, an anti-angiogenic TKI approved for the treatment of renal cell carcinoma. Axitinib was applied to the tumor−vessel interface model at 4 µM [17]. At this concentration, axitinib completely blocked cell mobilization from both the AR and the MCTS (Appendix A). This also included the cell types not affected by ZM323881. The more drastic effect could be attributed to the fact that axitinib also inhibits PDGFRs and cKIT.

## 4. Discussion

Although research initially focused more on angiogenic events, it is understood that both angiogenesis and vasculogenesis, or specifically postnatal vasculogenesis, contribute to neovessel formation under most pathological conditions [18,19,20,21,22]. Different tissues have been identified in the past three decades as an activatable resource of endothelial and/or endothelial progenitor cells, such as bone marrow, umbilical cord, and the adventitial layer of the vessel wall [4,23,24,25,26]. CD34^+^ vascular adventitia-resident stem and progenitor cells (VW-SPCs) have been shown to deliver both endothelial cells and peri-endothelial cells, such as pericytes and smooth muscle cells, that contribute to the stabilization of new vessels [8,27,28]. Furthermore, adventitia-derived CD44^+^ cells were reported to differentiate into pericytes that enwrapped the endothelial lining of new vessels from the outside, similar to pericytes in situ [29,30]. Beside vascular cells, VW-SPCs were shown to be able to generate macrophages independently from bone marrow [8,31,32,33]. However, until now, the role of these cells in tumor vascularization had not been studied in detail. The present results demonstrate that (i) in human tumor tissues, in situ CD34^+^ cells residing within the adventitial layer of pre-existing blood vessels are recruited from their niche in a tumor−vessel distance dependent manner; (ii) these cells infiltrate the tumor tissue as cell cords, forming capillary-like channels; (iii) in an aortic ring assay (ARA), VW-SPCs exert a great impact on the pattern of capillary-like sprouting and their morphogenesis, as well as on network formation, e.g., through vascular stabilization by enabling the incorporation of pericytes into the capillary wall; (iv) by mimicking the aforementioned pattern in human tumors in situ, 3D AR-MCTS-confrontation analyses in collagen gel as the matrix showed the mobilization of the adventitial CD34^+^ VW-SPCs and their recruitment to the tumor tissue, where they subsequently activate new vessel formation and thus tumor vascularization; and, (v) in contrast to classical angiogenesis provided by mature pre-existing endothelial cells, the postnatal vasculogenesis by adventitial VW-SPCs is not significantly suppressed by treatment with angiogenesis inhibitors like ZM323881.

Aortic ring assay (ARA) has been introduced in the field of vascular and angiogenesis research for several decades, and is still employed by many researchers as a tool to study the processes of neovascularization [34,35,36,37]. In addition to fully differentiated endothelial cells in the intimal layer and in various smaller supplying vessels (*vasa vasorum*), the vessel wall and the adventitial layer in particular has been shown to harbor various stem and progenitor cell populations (VW-SPCs) that can give rise to all cell types of mature vessels, such as endothelial cells, smooth muscle cells, and pericytes [8,12,13,38]. In order to evaluate the contribution of pre-existing mature endothelial cells versus those of VW-SPCs to new vessel formation, we established a mdARA by removing the adventitial layer prior to embedding into the collagen gel. Importantly, the protocol we established ensured that the removal of the adventitia did not result in any structural damage to the rest of the vessel wall. mdARA enabled examining the contribution of these VW-SPCs in detail and revealed that they strongly influence the vascular sprouting pattern in ARA. The presence of the adventitia resulted in the formation of more robust vessel sprouts organized in a more complex network. Moreover, our mdARA also enabled being able to differentially assess the effects of tumors on the mature endothelial cells of the vascular intima on one hand and the adventitial VW-SPCs on the other. The AR-MCTS confrontation assay showed that adventitial VW-SPCs are mobilized from their niche, directed to the MCTS as single cells and/or cell cords that (a) surround the tumor spheroid and (b) infiltrate the MCTS where they form vessel-like structures. This is in line with our initial findings that CD34^+^ cells from existing peritumoral blood vessels are mobilized to the tumor surrounding in bladder and breast cancer patients. Thus, our ARA model recapitulates this aspect of actual tumor progression. The presence of MCTS in close vicinity to ARs significantly alters the pattern of sprouting.

An unexpected observation of this study was that in the AR-MCTS confrontation assay, a part of tumor cells, especially under hypoxic conditions, formed cell cords reminicent of vasculogenic mimicry. These tumor cells also exhibited increased levels of CD34, an observation in line with previous reports [39,40,41]. Vasculogenic mimicry has been observed in various malignancies, such as breast cancer, lung cancer, prostate cancer, melanoma, glioblastoma, osteosarcoma, and hepatocellular carcinoma [42,43,44,45,46,47,48]. It has been implicated as a process important for tumor growth and thus, as a potential therapeutic target [49,50]. Our data suggest that a AR-MCTS confrontation model could serve as a platform to study this process ex vivo, and to better explore the role of this processes in tumor growth and metastasis. However, further studies are needed to gain better insight into the vasculogenic mimickry-like morphogenetic events of this assay.

Considering that not only large and mid-sized, but small blood vessels, harbor CD34^+^ cells within the outermost layer of their wall, adventitial VW-SPCs are omnipresent in vivo this vasculogenic potential might affect the efficacy of anti-angiogenic tumor therapy. The VEGF-R-2 inhibitor ZM323881 inhibited sprouting drastically when the adventitial layer in the ARA was removed. However, VW-SPCs seem to be able to initiate and maintain the sprouting process to some extent even under the inhibition of VEGF-R-2 signaling, as the effect was strongly diminished when the adventitial layer was left in place. This could help to explain the problems observed with anti-angiogenic therapy: development of resistance, continued-although dampened-vascularization under therapy, and the inability to completely starve tumors into regression, as it was originally envisioned in the development of those therapeutics [51,52,53,54,55]. The results from our mdARA indicate that, in particular, larger vessels that are co-opted by the tumor could continue to provide the tumor with new vessels for expansive growth and increasing the nutritional demands under VEGF-targeted therapy. The mobilized cells derived from VW-SPCs of those large vessels could supply endothelial cells with additional angiogenic factors. Macrophages that we have shown to also be expanded from the adventitial layer secrete a range of angiogenic factors, including VEGF-A, FGF-2, and SDF-1 [56,57,58,59]. In addition, the VW-SPCs act as a reservoir for pericytes, that can directly incorporate into the new vessel sprouts as we have shown. Pericytes stabilize endothelial cells and make them less sensitive to VEGF inhibition [60,61,62].

Taken together, the present results demonstrate that vascular adventitia-resident VW-SPCs serve as a local source for tumor vascularization. As demonstrated here by studies on human tumors, these cells can be mobilized from the vascular adventitia and recruited to tumor tissue, probably by paracrine mechanisms, e.g., VEGF, which is released by adventitia-derived macrophages, if the tumor is in close proximity to the adventitial layer of the pre-existing blood vessels. These data can be confirmed by the results obtained from the ex vivo AR-MCTS confrontation assay. Moreover, in contrast to the non-tumor spheroids, e.g., neural spheroids that were directly penetrated by capillary sprouts from the ARs, in cases of ARs confrontation with tumor spheroids, the capillary sprouts stopped at the border of the tumor spheroids and, starting with culture day 6, they regressed to some extent instead of further growth. Moreover, individual aortic wall-derived VW-SPCs penetrated into the tumor spheroids and formed new vessels within the tumor spheroids, while no direct penetration of capillary-like sprouts into the tumor spheroids could be observed. Of note, this behavior is very close to what we observed by CD34 immunostaining on human urinary bladder and breast cancer tissues, indicating that our vessel wall−MCTS confrontation model is suitable for analyzing the contribution of vessel wall-derived VW-SPCs to tumor vascularization at a more differential level, as it was possible, yet, e.g., separating the role of mature endothelial cells from the vascular wall progenitors. By this, the presented model also enables differential testing of the effects of angiogenesis inhibitors under ex vivo conditions where not only vascular cell types, e.g., endothelial cells, pericytes, and smooth muscle cells are present, but also vascular wall-derived macrophages that are known to essentially contribute to tumor growth and metastasis. Thus, the un-appreciated role of VW-SPCs and their cellular and humoral derivatives could explain to some extent the failure or lower efficacy of the anti-angiogenic therapy in clinical trials and the treatment of tumor patients.

## Figures and Tables

**Figure 1 cells-10-01719-f001:**
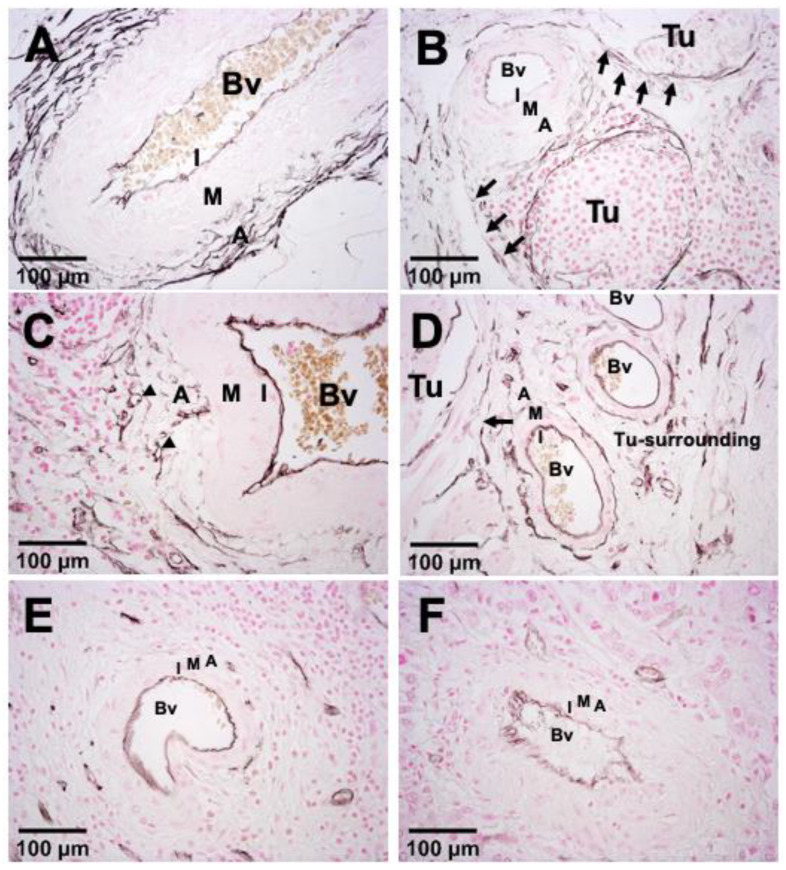
Mobilization of adventitia-resident CD34^+^ cells in tumors. Stepwise mobilization of adventitial CD34^+^ cells in human bladder cancer (**A**–**F**). Artery in the normal, non-transformed tissue surrounding the tumor: a clear, stable CD34^+^ zone is discernible in the adventitia (**A**). Blood vessel (Bv) in close vicinity to the tumor (Tu): CD34^+^ cells are partially detached from the adventitia, migrating towards the tumor area (arrows), forming small vessels (arrow heads) (**B**,**C**). Tumor surrounding area without invasion of tumor cells: CD34^+^ cells are still detectable in the entire adventitia of the blood vessels (**D**). Blood vessels that are fully enclosed by the tumor: the adventitia is depleted of CD34^+^ cells (**E**,**F**). Abbreviations: Bv—blood vessel; Tu—tumor; I—vascular intima; M—vascular media; A—vascular adventitia.

**Figure 2 cells-10-01719-f002:**
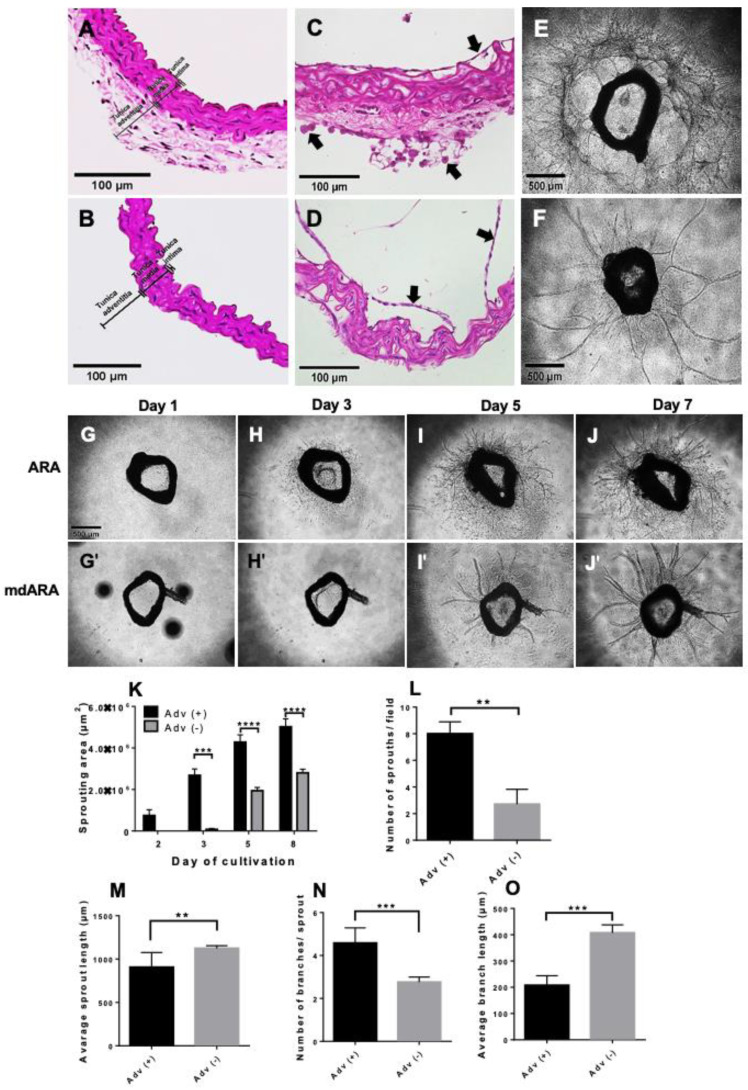
Removal of the adventitia delays sprouting and modulates the sprouting pattern in ARA. H&E stained paraffin sections of freshly isolated mouse aorta embedded with (**A**) and without adventitia (**B**). ARA and mdARA after 7 days of cultivation: sprouting originated from the intima, but also from the adventitia (arrows) (**C**), in mdARA only from the intima (**D**). Phase contrast images of ARA (**E**) and mdARA (**F**) after 7 days of cultivation. Time course of sprouting and branching in ARA (**G**–**J**) and mdARA (**G’**–**J’**). In the absence of adventitial cells, both processes are clearly delayed. Removal of the adventitia results in longer, less branched sprouts. Quantification of the sprouting area (**K**), number of sprouts/field (**L**), average sprout length (**M**), number of branches/sprout (**N**), and average branch length (**O**) in ARA and mdARA on day 8. Error bars indicate SD. ** *p* ≤ 0.01, *** *p* ≤ 0.001, **** *p* ≤ 0.0001.

**Figure 3 cells-10-01719-f003:**
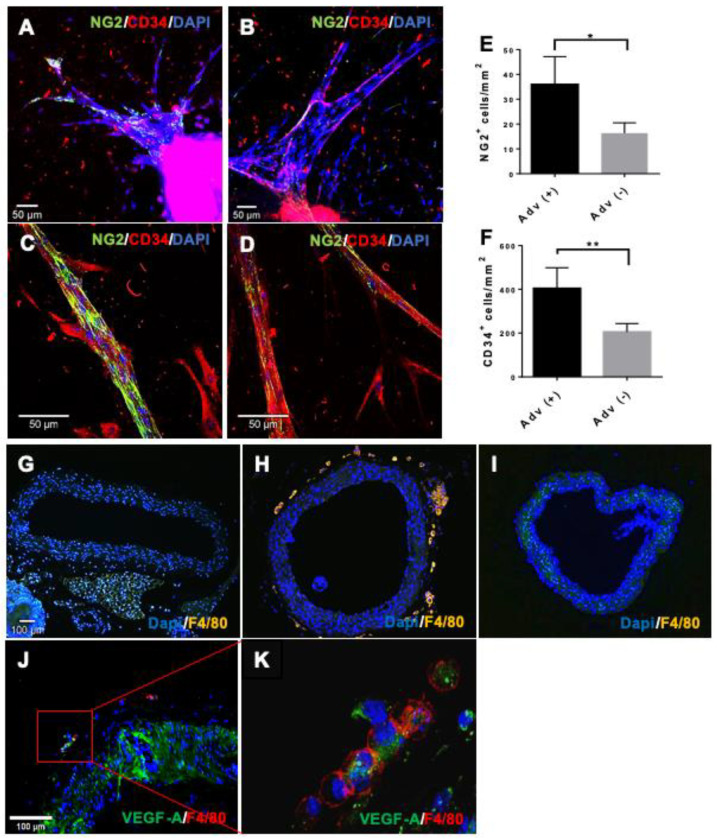
Vascular adventitia harbors sprouting-relevant cell types. Whole-mount staining of ARA and mdARA on day 7, for NG2 (pericytes (green)) and CD34 (endothelial and hematopoietic progenitor cells (red)). Confocal images: (**A**,**C**) in ARA more NG2^+^ cells are visible than in mdARA (**B**,**D**). Quantification of NG2^+^ and CD34^+^ cells from A and B (**E**,**F**). Staining for F4/80^+^ cells (yellow) in freshly isolated aorta (**G**). ARA (**H**) and mdARA (**I**) after 7 days of cultivation. F4/80^+^ cells are absent in the freshly isolated aorta, but appear within the adventitia during cultivation. A subset of F4/80^+^ cells stain positive for VEGF-A (**J**,**K**). ARA (aorta from Cx3cr1 CreER+/mTmG+/- reporter mice): fluorescent marked cells in non-fixed live images (**L**) and DAB immunostaining on paraffin section of ARA demonstrated the presence of F4/80^+^ cells in the regions of branching points (arrows) of capillary sprouts (**M**–**O**). H&E staining of freshly isolated adventitia shows cells (arrows) in an inactive state (**P**). After cultivation in collagen for 7 days, capillary-like sprouts (arrows) become visible (**R**) with observable lumen (Lu) formation (**S**). Capillary-like sprouts are composed of CD34^+^ and NG2^+^ cells (**T**). Along sprouts, F4/80^+^ cells are visible (**U**,**V**). Error bars indicate SD. * *p* ≤ 0.05, ** *p* ≤ 0.01.

**Figure 4 cells-10-01719-f004:**
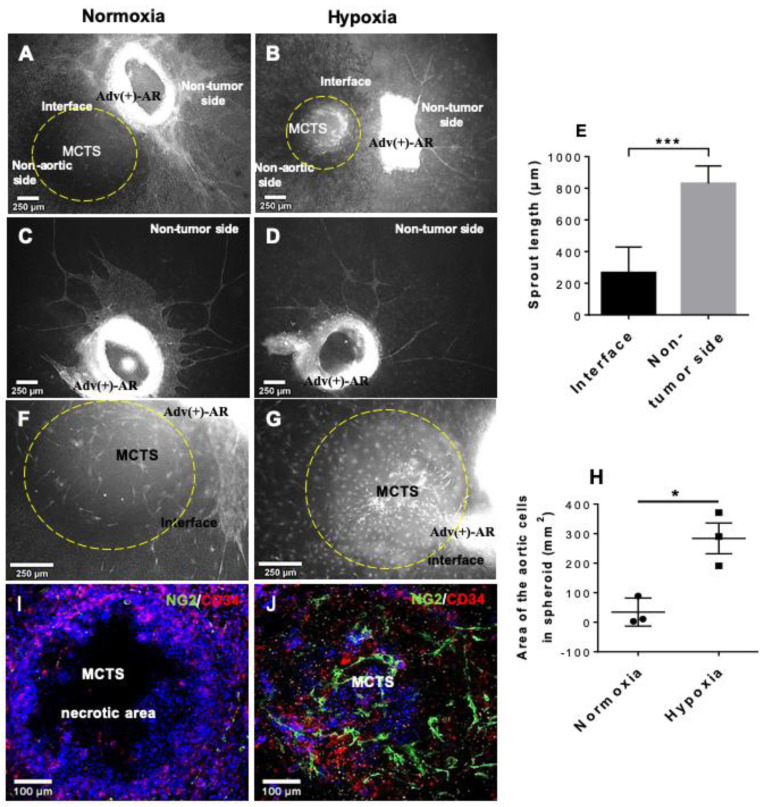
3D MCTS confrontation model with ARs from GFP^+^ mouse aorta (AR-MCTS). MCTS confrontation with ARs cultivated for 11 days in collagen gel under normoxic (**A**) or hypoxic conditions (**B**). Aortic cells were GFP marked. At the non-tumor side, angiogenic sprouting occurred under normoxia and hypoxia (**C**,**D**), whereas in the interface, sprouting was only observed under hypoxia (**B**). Quantification of the sprouts (**E**). Sprouting towards the MCTS is observable after 11 days of confrontational cultivation under normoxic (**F**–**I**), and hypoxic conditions (**G**–**J**). Under normoxia, few NG2^+^ aortic cells only scarcely surrounded the MCTS without penetrating the spheroid (**I**). Under hypoxia, more NG2^+^ cells were mobilized from the AR, forming a dense network within the MCTS (**J**). Quantification from F and G (**H**). Error bars indicate SD. * *p* ≤ 0.05, *** *p* ≤ 0.001.

**Figure 5 cells-10-01719-f005:**
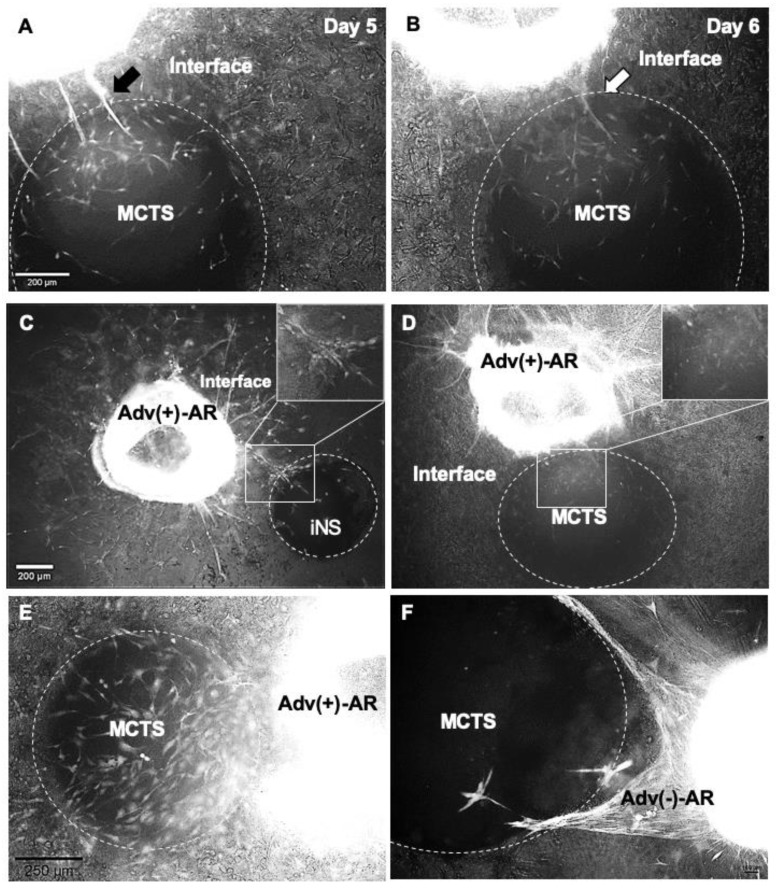
Dependence of sprouting patterns on cell types of the spheroid and VW structure. In the tumor−vessel interface, transient capillary sprouts were built by day 5 (**A**), but started to disappear at day 6 at the border to MCTS (**B**). AR co-embedded with iNS (**C**) or MCTS (**D**) after 8 days, under normoxic conditions. If Adv(+)-ARs were co-embedded with iNS, capillary sprouts directly penetrated the spheroids (**C**). By co-embedding with MCTS, only discrete, mobilized cells can be observed in the interface, while capillary-like sprout formation is absent (**D**). Co-embedding of Adv(+)-AR (**E**) or Adv(−)-AR (**F**) with MCTS after 8 days, under hypoxic conditions. Abundant GFP^+^ aortic cells of Adv(+)-AR migrated towards the MCTS (**E**). In the absence of an adventitia and thus, also the absence of VW-SPCs, the migration of single cells is reduced, but cell cords form around the MCTS (**F**).

**Figure 6 cells-10-01719-f006:**
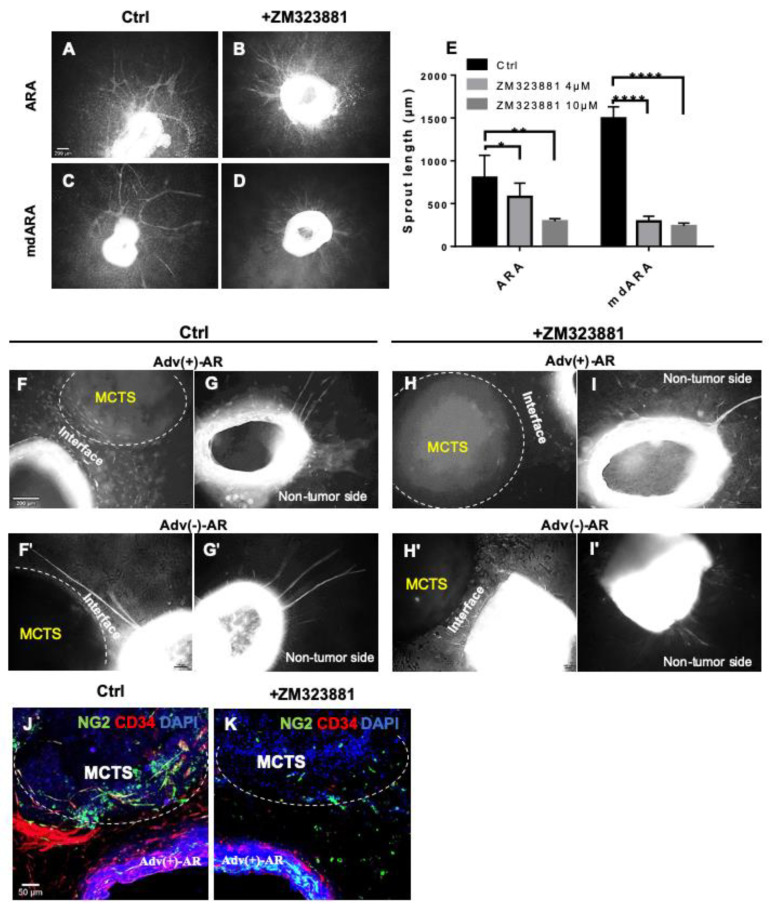
Vascular adventitial cells affect response to VEGF-R inhibition. ZM323881 reduces vascular sprouting less efficiently in ARA with the adventitia in place (**A**,**B**) than in mdARA (**C**,**D**). The quantification of sprouting (**E**). Treatment with ZM323881 reduced the migration of cells towards the MCTS in the Adv(+) experiments (**F** vs. **H**), but had little effect on sprouting at the non-tumor side (**G** vs. **I**). In the Adv(-) experiments, ZM323881 completely blocked capillary sprouting (**F’**–**H’** versus **G’**–**I’**). Whole mount staining of CD34^+^ cell in red and NG2^+^ cells in green (**J**,**K**), under hypoxia, at day 14. Error bars indicate SD. * *p* ≤ 0.05, ** *p* ≤ 0.01, **** *p* ≤ 0.0001.

## Data Availability

The data presented in this study are available on request from the corresponding authors.

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
