# Peer review of "Contribution of Adventitia-Derived Stem and Progenitor Cells to New Vessel Formation in Tumors"

_cells, 2021, doi:10.3390/cells10071719_

Round 1
Reviewer 1 Report
The study by Upcin et al., presents a study on the role of adventitia-derived stem and progenitor cells in neovessel formation. The authors employ in vivo and an ex vivo co-culture model of multi-cellular tumor spheroids (MCTS) with aortic ring vessel segments. The findings presented appear to be interesting, robust, and of meaningful interest to the intended readership. In particular, the finding that neovascularization by VW-SPCs is largely unaffected by the anti-VEGFR-2 therapy is particularly compelling as it points to a possible resistance mechanism, which has troubled anti-angiogenesis therapies on oncology (as indicated by the authors). Based on the strength of the presented findings, I recommend publication with a few minor comments.
- Although the authors control the ambient O2 conditions for the normoxia versus hypoxia studies, it is also recommended that hypoxia is in situ confirmed with a hypoxia stain (e.g. pimonidazole)
- Some of the scale bar markings are not present or not legible (e.g. Figures 6A 5F)
- The manuscript indicates that ZM 323881 (anti-VEGFR-2) was applied at 4 μM and 10 μM. However, the basis for these concentrations tested is unclear and merits the necessary citation.
- In some instance, ZM 323881 has a comma between the second 3 and first 8 (e.g. Page 19, line 491)
Author Response
Response to the reviewers:
First, we would like to thank the reviewers for the time and effort they dedicated into reviewing our work. Thanks to your supportive suggestions were able to significantly improve our manuscript. Please find below in detail our answers to your questions and the description how we addressed your suggestions for improvement.
Kind regards,
The study by Upcin et al., presents a study on the role of adventitia-derived stem and progenitor cells in neovessel formation. The authors employ in vivo and an ex vivo co-culture model of multi-cellular tumor spheroids (MCTS) with aortic ring vessel segments. The findings presented appear to be interesting, robust, and of meaningful interest to the intended readership. In particular, the finding that neovascularization by VW-SPCs is largely unaffected by the anti-VEGFR-2 therapy is particularly compelling as it points to a possible resistance mechanism, which has troubled anti-angiogenesis therapies on oncology (as indicated by the authors). Based on the strength of the presented findings, I recommend publication with a few minor comments.
- Although the authors control the ambient O2 conditions for the normoxia versus hypoxia studies, it is also recommended that hypoxia is in situ confirmed with a hypoxia stain (e.g. pimonidazole)
We performed staining for carbonic anhydrase IX (CAIX) sections of MCTS+ARs that were cultivated under normoxic and hypoxic conditions (Supl. Fig 3).
- Some of the scale bar markings are not present or not legible (e.g. Figures 6A 5F)
We want to apologize for the inconvenienced caused by the illegible markings. We increased the type size.
- The manuscript indicates that ZM 323881 (anti-VEGFR-2) was applied at 4 μM and 10 μ However, the basis for these concentrations tested is unclear and merits the necessary citation.
The reviewer is of course correct. We have added the citations in the text.
- In some instance, ZM 323881 has a comma between the second 3 and first 8 (e.g. Page 19, line 491)
We corrected these inconsistencies and used the spelling ZM323881 throughout the text.
Reviewer 2 Report
The manuscript “Contribution of adventitia-derived stem and progenitor cells to new vessel formation in tumors” by Upcin et al. describes the importance of vascular adventitia resident cells recruitment in tumor vasculogenesis.
The paper is clearly written, and data presented is convincing and supported by adequate images. However, the manuscript requires additional data and information to be included, together with some text corrections.
My major concern is about the use of ZM323881 for functional experiments. Why, among all angiogenesis inhibitors used to treat different types of cancer, did author choose ZM323881? Can you provide some additional information about this inhibitor (i.e. references to clinical and preclinical previous studies)? Additional experiments should be performed, in order to exclude a possible effect of ZM323881 on cell proliferation or apoptosis. Moreover, the use of other classical anti-angiogenic drugs acting on VEGFRs should be added for comparison, and would increase the importance of presented data.
Minor points:
- Materials and methods.
- Lines 92-101 contains information to be included in discussion rather than in materials section.
- Line 106: please explain the abbreviation Adv(+)-ARs and Adv(-)-ARs .
- Line 112 and line 139: Please explain how hypoxic environment was created. Did author use a hypoxic chamber or an hypoxic incubator? Were cells maintained 8-11 days continuously under hypoxic conditions?
- Lines 138, 157, 161,163: typing and grammar errors should be corrected.
- Figure 2: please define the abbreviation mdARA, and add a scale bar to micrographs H-I-J-G’-H’-I’-J’.
- Figure 3: Scale bar in micrographs H,I,K,L,R,S,U,V is missing
- Figure 5: Scale bar in micrographs B,D,F is missing. Scale bars are not consistent (E differs from A and C).
- Figure 6: please add a clear scale bar.
- Author contributions: Lines 546-547 should be removed.
Author Response
Response to the reviewers:
First, we would like to thank the reviewers for the time and effort they dedicated into reviewing our work. Thanks to your supportive suggestions were able to significantly improve our manuscript. Please find below in detail our answers to your questions and the description how we addressed your suggestions for improvement.
Kind regards,
Comments and Suggestions for Authors
The manuscript “Contribution of adventitia-derived stem and progenitor cells to new vessel formation in tumors” by Upcin et al. describes the importance of vascular adventitia resident cells recruitment in tumor vasculogenesis.
The paper is clearly written, and data presented is convincing and supported by adequate images. However, the manuscript requires additional data and information to be included, together with some text corrections.
My major concern is about the use of ZM323881 for functional experiments. Why, among all angiogenesis inhibitors used to treat different types of cancer, did author choose ZM323881? Can you provide some additional information about this inhibitor (i.e. references to clinical and preclinical previous studies)?
We used ZM323881 as it is in contrast to other VEGFR2 inhibiting TKIs very selective for VEGFR2 inhibitor and has almost no activity on VEGFR1, PDGFRβ, FGFR1, EGFR and ErbB2. See for example Whittles et al. (2002), Microcirculation or the data compiled by Selleck Chemicals (Houston, TX, https://www.selleckchem.com/products/zm-323881-hcl.html). This allowed us to limit the observed effects to the VEGF-A/R2 axis.
To address the reviewer’s concern we repeated the experiments with axintinib (Supl. Fig. 4). As expected the results were similar but more extensive compared to ZM323881. In these additional experiments it is clear that additional cell populations are affected, as axitinib also targets PDGFRb/a (and cKit among others). The clinically approved VEGF-R2 inhibitors are generally more promiscuous than ZM323881.
Additional experiments should be performed, in order to exclude a possible effect of ZM323881 on cell proliferation or apoptosis. Moreover, the use of other classical anti-angiogenic drugs acting on VEGFRs should be added for comparison, and would increase the importance of presented data.
We have to apologize, but we are not sure about the reviewer’s concerns in this instant. ZM323881 is a potent VEGFR-2 inhibitor, we explicitly utilized this drug to block VEGFR-2 signaling. VEGFR-2 signaling is essential for proliferation and maintenance of endothelial cells. Thus, we would absolutely expect to have effects of ZM323881 on cell proliferation and apoptosis but the study of such effects is not the focus of the current manuscript.
As mentioned before we have repeated core experiments with clinically approved TKI axitinib. We absolutely agree with the reviewer that these additional data increase the impact of our findings.
Minor points:
- Materials and methods.
- Lines 92-101 contains information to be included in discussion rather than in materials section.
We agree that the paragraph in its elaborateness was written more like a part of the discussion. However, as it only contains information essential for reproducing our experiments, but not necessary for the general understanding of the presented results, we decided to shortened the passage and left it in the methods section.
- Line 106: please explain the abbreviation Adv(+)-ARs and Adv(-)-ARs .
- Line 112 and line 139: Please explain how hypoxic environment was created. Did author use a hypoxic chamber or an hypoxic incubator? Were cells maintained 8-11 days continuously under hypoxic conditions?
We used a hypoxic incubator. We clarified the exact procedure in the methods section.
- Lines 138, 157, 161,163: typing and grammar errors should be corrected.
Thanks a lot for pointing these mistakes out. We have corrected them.
- Figure 2: please define the abbreviation mdARA, and add a scale bar to micrographs H-I-J-G’-H’-I’-J’.
- Figure 3: Scale bar in micrographs H,I,K,L,R,S,U,V is missing
- Figure 5: Scale bar in micrographs B,D,F is missing. Scale bars are not consistent (E differs from A and C).
- Figure 6: please add a clear scale bar.
All scales bars have been added/improved in clarity.
- Author contributions: Lines 546-547 should be removed.
We have removed these lines. Thank you for finding the mistake.
Round 2
Reviewer 2 Report
The criticisms have been answered, no other comment